# Impacts of Verticillium Wilt on Photosynthesis Rate, Lint Production, and Fiber Quality of Greenhouse-Grown Cotton (*Gossypium hirsutum*)

**DOI:** 10.3390/plants9070857

**Published:** 2020-07-07

**Authors:** Addissu G. Ayele, Terry A. Wheeler, Jane K. Dever

**Affiliations:** Texas A&M AgriLife Research, 1102 E. Drew St., Lubbock, TX 79403-6603, USA; addissu.ayele@ag.tamu.edu (A.G.A.); twheeler@ag.tamu.edu (T.A.W.)

**Keywords:** disease resistance, *Gossypium hirsutum*, *Verticillium dahliae*

## Abstract

Verticillium wilt, caused by *Verticillium dahliae* Kleb., leads to significant losses in cotton yield and fiber quality worldwide. To investigate Verticillium wilt impact on photosynthesis rate, yield, and fiber quality, six upland cotton genotypes, namely Verticillium susceptible (DP 1612 B2XF) and partially resistant (FM 2484B2F) commercial cultivars and four breeding lines, were grown to maturity under greenhouse conditions in soil either infested or not infested with *V. dahliae* microsclerotia. Photosynthetic rate, lint, and seed yield were all higher (*p* < 0.05) for FM 2484B2F than DP 1612 B2XF when infected with *V. dahliae*. When comparing healthy (H) to Verticillium wilt (VW) affected plants, fiber properties were greatly impacted. Micronaire decreased from 5.0 (H) to 3.6 (VW) with DP 1612 B2XF and 4.4 (H) to 4.1 (VW) with FM 2484B2F. The maturity ratio decreased from 0.90 (H) to 0.83 (VW) for DP 1612 B2XF and was unchanged for FM 2484B2F (0.90). Fiber properties such as short fiber content, nep count, fineness, and immature fiber content were also significantly affected under Verticillium wilt pressure. With Verticillium wilt affected plants, lines 16-13-601V and 17-17-206V performed similarly to FM 2484B2F for photosynthetic rate, yield, and all fiber properties measured. When selecting for improved cultivars in the presence of Verticillium wilt, it is important to select for relatively unchanged fiber properties under disease pressure in addition to reduced disease severity and increased yield.

## 1. Introduction

Verticillium wilt in cotton is caused by *Verticillium dahliae* Kleb., a soil-borne fungus. This disease causes substantial losses in cotton worldwide [1]. The fungus produces infectious hyphae that emerge from microsclerotia (multicellular, long-lived structures in the soil) and colonize plant roots. The fungus enters the root vascular tissue and then spreads upwards in the vascular system [2]. Symptoms of Verticillium wilt include stunting, wilting, chlorosis, foliar desiccation, defoliation, and plant death. The stress imposed by Verticillium wilt reduces the rate of photosynthesis [3,4,5,6,7]. Water stress is often associated with wilt pathogens that colonize the xylem [2,7,8]. The authors of [9] reported that the combined effect of chlorosis, water stress, and stomatal closure could accelerate the reduction in photosynthesis rate of leaves that exhibited disease symptoms. Therefore, there is a substantial loss of lint yield and fiber quality associated with Verticillium wilt of cotton [10,11], which may be caused by the reduction in photosynthesis and increased resistance to hydraulic conductance in the xylem.

Management of Verticillium wilt in most crops is difficult. Soil fumigation has been practiced effectively in some high-value crops like strawberries [12]. However, cotton has insufficient value for soil fumigation to be cost-effective. Crop rotation has been unsuccessful for the control of Verticillium wilt [13,14], likely due to the long-lived nature of microsclerotia [15]. Although some biological remedies, including the incorporation of broccoli into the soil [16], biological disinfestation using organic matter [17], and an integrated management strategy for Verticillium wilt [18], are successful at reducing Verticillium wilt, the effectiveness and economic feasibility of these techniques remain ambiguous. The most economically feasible and environmentally friendly method of managing the pathogen is the use of cotton cultivars with at least partial resistance to Verticillium wilt [19,20,21,22].

Verticillium wilt damages the quality of cotton fiber, including length, strength, and grade [23]. Yarn spun from wilted plants was inferior in appearance, and Verticillium wilt affected plants produced more neps and resulted in greater manufacturing waste [23]. Verticillium wilt may reduce deposition and reorganization of cellulose molecules in cotton fiber. This could impact fiber yield and fiber properties, including micronaire, fiber maturity, short fiber content, and immature fiber content, as these are all related to cellulose deposition and reorganization in cotton fiber development. The objective of this project was to compare the rate of photosynthesis, yield, and fiber properties in the absence and presence of Verticillium wilt on germplasm ranging from very susceptible to partially resistant.

## 2. Results

### 2.1. Disease Ratings

The first symptoms associated with Verticillium wilt appeared on average between 44 and 67 days after planting (DAP) for the six genotypes (Table 1). The partially resistant commercial cultivar FM 2484B2F and breeding line 16-13-203V developed initial symptoms at 47 and 44 DAP, respectively, and line 16-13-601V was the last to develop symptoms at 67 DAP. All other genotypes were intermediate in symptom initiation. Disease severity was highest for the susceptible commercial cultivar DP 1612 B2XF (3.5) and 16-13-203V (3.7) (Table 1). All the other breeding lines and FM 2484B2F had less severe disease symptoms (Table 1). Vascular (stem) discoloration at harvest was worse for DP 1612 B2XF (2.4) than for 16-13-601V (1.0), 17-17-206V (1.4), 17-17-606V (1.5) and FM 2484B2F (1.0) (Table 1). Disease severity and vascular stem discoloration suggested that FM 2484B2F, 16-13-601V, and 17-17-206V were more resistant than DP 1612 B2XF and 16-13-203V. Earliness of symptom initiation was not predictive of disease severity or stem discoloration.

### 2.2. Photosynthesis

Significant genotype × inoculum (with and without *V. dahliae*) and genotype × DAP interactions were observed for photosynthesis and transpiration rates (Table 2), suggesting that some genotype(s) performed well in the absence of disease, while the same genotype(s) performed poorly under Verticillium wilt pressure. Conversely, genotype × DAP interactions for intercellular carbon dioxide concentration were not significant. Growth stages characterized by DAP showed significant differences for all physiological parameters considered. Photosynthetic rate, intercellular carbon dioxide concentration (Ci), and transpiration rate showed significant effects from the presence/absence of *V. dahliae*.

Although Verticillium wilt symptoms occurred before flowering, the expression of the disease was greatest after flowering. In the first 50 DAP, no differences were observed in the photosynthesis rate between plants with Verticillium wilt and healthy plants (Figure 1). However, a significant reduction in the photosynthetic rate was observed from the early flowering stage (57 DAP) through boll set (86 DAP). The range between 86 and 105 DAP overlapped with the peak boll-setting period and the first 25% of the boll-cracking stage. After 86 days, leaf chlorophyll began degrading, and the photosynthesis rate of both healthy and diseased plants declined.

In the absence of disease, all cotton genotypes had a similar rate of photosynthesis. However, differences between genotypes were seen in the presence of disease (Table 2). The highest percent loss in photosynthesis rate occurred with DP 1612 B2XF and 16-13-203V (Figure 2). Lines 17-17-206V, 16-13-601V, and 17-17-606V had a rate of photosynthesis that were similar to FM 2484B2F. The loss in photosynthesis rate mirrored the differences seen in damage severity ratings and vascular necrosis.

### 2.3. Yield and Fiber Quality

There was a drastic change in the lint and seed weights between healthy plants and plants with Verticillium wilt (Figure 3). Lint weight decreased by 32% in diseased 16-13-203V plants and by 46% in diseased DP 1612 B2XF plants compared to their healthy counterparts. Genotypes whose lint yield was less adversely affected in diseased plants included 16-13-601V (15% reduction), 17-17-206V (22% reduction), and FM 2484B2F (24% reduction). The effect of Verticillium wilt on seed production per plant showed similar trends as those seen for lint yield (Figure 3). The lower percent lint and seed yield loss for 16-13-601V and 17-17-206V suggest tolerance of these two genotypes to Verticillium wilt.

There was a genotype × inoculum interaction with micronaire measured by high volume instrument (HVI) and length by number, short fiber content, fineness, immature fiber content, and maturity ratio measured by the advanced fiber information system (AFIS). However, Verticillium wilt did not affect genotype × inoculum interaction with HVI-measured fiber length, length uniformity, strength, and elongation. All genotypes grown in the absence of disease produced fibers with a higher micronaire (Table 3). Some genotypes (17-17-606V, 16-13-203V, 16-13-203V, and DP 1612 B2XF) with Verticillium wilt produced fibers with relatively low micronaire. The partially resistant cultivar (FM 2484B2F) and advanced breeding lines 17-17-206V and 16-13-601V produced similar micronaire values under Verticillium wilt pressure. The trend in short fiber content followed micronaire. The highest fiber breakage was recorded for DP 1612 B2XF and 16-13-203V breeding lines (Table 3). With diseased plants, the average short fiber content was significantly lower for 16-13-601V, 17-17-606V, and 17-17-206V than for DP 1612 B2XF. Interestingly, the lowest average short fiber content was recorded for DP 1612 B2XF in the absence of Verticillium wilt, while the highest average short fiber content was recorded for the same cultivar in the presence of Verticillium wilt. All genotypes with Verticillium wilt produced significantly higher nep count and short fiber content compared to their healthy counterparts. There was no significant difference between genotypes for nep count in healthy plants. DP 1612 B2XF and 16-13-203V produced relatively high percent immature fiber content, explained by higher nep count and short fiber content (Table 3). FM 2484B2F with Verticillium wilt produced mature fibers and had low immature and short fiber contents. Breeding lines 16-13-601V and 17-17-206V followed the trends of fiber quality produced by FM 2484B2F (Table 3).

For healthy plants, all genotypes produced the nearly perfect fiber length distribution that is normally desired by the textile industry (Figure 4). In the healthy plants, genotypes produced a minimum number of short fibers with well-defined long fiber peaks. In contrast, the frequency of short fibers created in most cotton genotypes with Verticillium wilt was high, possibly due to low fiber maturity. Fibers produced by DP 1612 B2XF and 16-13-203V were seriously damaged by Verticillium wilt. These types of fibers have limited use in yarn processing, as the length distribution reveals that such fibers are below spinnable quality. When healthy and diseased genotypes were compared, 16-13-601V produced a minimum number of short fibers and well-defined long fiber frequencies and thus can be considered potentially spinnable in the current textile industry.

## 3. Discussion

Verticillium wilt restricts the movement of water and nutrients within the vascular system of cotton and drastically limits crop productivity. We have observed that the photosynthesis rate of each cotton genotype was reduced by Verticillium wilt, although the response mechanisms which affect photosynthesis rate and other related physiological traits may depend on genotype. The disease was less severe, including necrosis and presumably blockage of the xylem, depending on genotype. Cotton genotypes show different responses to Verticillium wilt [9]. In this study, some genotypes showed high susceptibility while other genotypes showed partial resistance when planted into *V. dahliae* infested soil. Resistance mechanisms to *V. dahliae* may include cell wall modifications, extracellular enzymes, transcription factors, pattern recognition receptors, jasmonic acid, salicylic acid, or ethylene-related signal transduction pathways [24].

The magnitude of damage caused by Verticillium wilt to cotton lint and seed yield depends on the phenotypical stage of the plant when first foliar symptoms occur in the crop [25]. Our results did not necessarily agree with this observation. FM 2484B2F had early symptom initiation of Verticillium wilt but is one of the most resistant commercial cultivars to this disease [22]. However, 16-13-203V had the earliest symptom initiation for Verticillium wilt but was as susceptible to this disease as DP 1612 B2XF. Minimal season-long disease severity combined with low yield loss in Verticillium wilt fields are both important indicators for selecting Verticillium wilt resistant germplasm [22].

Verticillium wilt essentially causes premature senescence in plants, and this early termination of productivity may reduce deposition and organization of cellulose molecules in cotton fiber. Cellulose deposition and organization of the secondary cell wall is an important structural change that occurs during cotton fiber development. The composition of mature cotton fiber is 88 to 97% cellulose [26]. Any external factors that affect the structural change of the cell wall could impact fiber properties such as micronaire, fiber maturity, short fiber content, and other fiber-maturity-related properties. It has been reported that *Verticillium* spp. induce cell-wall-degrading enzymes and phytotoxins [27]. When *V. dahliae* infects a cotton plant, carbohydrate-active (CAZymes) protein molecules that participate in pectin and cellulose degradation pathways significantly activate the corresponding transcription levels of several genes encoding plant cell wall degradation enzymes [28] that cause inferior fiber quality.

The initiation of secondary cell wall synthesis depends on genetic differences in cotton lines [26]. The impact of Verticillium wilt on the secondary cell wall development may have differed between genotypes. Genotypes considered in this study showed variable fiber properties, particularly for micronaire, maturity, fineness, immature fiber content, and short fiber content. In cotton fiber development, the period between 16 to 21 days post-anthesis (DPA) marks a developmental change from primary to secondary cell wall synthesis [29]. The highest rate of cellulose synthesis commenced from 24 to 25 DPA after the cessation of the elongation phase [26,28]. It appears that severe disease symptoms caused by Verticillium wilt during early cell wall synthesis drastically reduced maturity and fineness of the fiber, which led to low micronaire of susceptible cotton lines. Micronaire is the combination of fineness and maturity, directly proportional to the measure of airflow between cotton fibers. More immature fibers may be created when Verticillium wilt is present.

Previous studies indicated that immature fibers break during processing [30,31,32,33]. In this study, it appears that the effect of Verticillium wilt results in poor development of secondary cell walls, rendering them weak and with a propensity to break during mechanical processing and create higher short fiber content. In the presence of high immature fiber content, the entanglement of fibers is expected. The results of this study show that upland cotton genotypes (DP 1612 B2XF and 16-13-203V) produced a high percentage of immature fiber, leading to the entanglement of fibers. Conversely, some germplasm with Verticillium wilt exhibited higher micronaire, lower short fiber content, low nep count, low immature fiber content, and relatively more mature fibers. The high variabilities observed between diseased and healthy upland cotton fiber quality suggests the possibility of a large gap to be filled through improving cotton against Verticillium wilt.

When the complete fiber length distributions were compared among genotypes, the fiber breakage of susceptible cotton lines was very high under disease stress compared to that of healthy plants. This study demonstrated that the spinnability of the fiber into quality yarn was negatively impacted by Verticillium wilt in susceptible cotton germplasm. However, more resistant germplasm like 16-13-601V was able to maintain a consistent fiber length distribution with a minimum number of short fibers and well-defined long fiber peaks, even with Verticillium wilt. When breeding for germplasm with resistance to Verticillium wilt, it is important to include reduced disease severity, high yield, and good fiber quality traits.

## 4. Materials and Methods

### 4.1. Germplasm

The genotypes tested were four unreleased cotton (*Gossypium hirsutum*) breeding lines (16-13-203V, 16-13-601V, 17-17-206V, and 17-17-606V) developed at Texas A&M AgriLife Research in Lubbock, TX. These breeding lines were evaluated in the field for Verticillium wilt resistance. The advanced breeding lines Deltapine brand DP 1612 B2XF, PVP 201600049, (Verticillium wilt susceptible commercial cultivar) and FiberMax brand FM 2484B2F, PVP 201200291, (Verticillium wilt partially resistant commercial cultivar) were selected for the greenhouse experiment because they demonstrated partial resistance in an open environment [22].

### 4.2. Fungal Preparation and Soil Inoculation

Microsclerotia of *V. dahliae* (defoliating type isolate) were produced on a minimal medium overlaid with cellophane [34]. The microsclerotia were harvested from the cellophane six weeks after inoculation and washed through a sieve with a 230-µm pore opening stacked over a sieve with a 37-µm pore opening. The contents on the 37-µm sieve were mixed into dried and sieved soil and used as the “hot mix”. The soil was sieved with a 0.635 cm mesh sieve. The soil used was an Acuff loamfine-loamy mixed thermic Aridic Paleustolls (50% sand, 21% silt, 29% clay) with a pH of 7.8 and 0.5% organic matter. Soil which had no *V. dahliae* microsclerotia before inoculation was air-dried, sieved, and mixed, either with microsclerotia of *V. dahliae* or mixed with no microsclerotia, for 180 seconds in a twin-shell blender (Model 1-CU-FT twin-shell blender with intensifier bar; Patterson-Kelley, East Stroudsburg, PA). For calibration of the inoculum, 250 cm^3^ of the hot mix was mixed in a twin-shell blender with 20 L of soil, and then three subsamples (10 cm^3^ soil/subsample) were assayed by dilution plating on Sorensen’s NP-10 media, with the pH adjusted to 5.5 [35]. The amount of “hot mix” in each experiment was then adjusted to reach the desired density, and the final microsclerotia density added to pots in each experiment was determined from the same dilution plating methodology as the calibration step. The final density after mixing was 808 microsclerotia/g soil.

Each pot (39.6 cm tall, 30.0 cm diameter at top, 25.8 cm diameter at the bottom, 22.2 L volume) held 19 L of soil. The mixed soil was placed in the center of the pot in a tube (8 cm diameter) (Figure 5). Nonmixed soil with no natural infestation of *V. dahliae* was then filled around the tube. The tube was then pulled out, leaving the mixed soil in the center of the container. The outer layer of soil was not mixed to provide better soil structure for air and water movement. Four seeds were planted into the center and thinned to one plant/pot after emergence. The seed was planted within a day of adding the inoculated soil to pots. Pots were watered uniformly as needed until plants were established and then watered at 2- to 3-day intervals after that time. Soil in the outer part of the pot (not mixed) had a granular slow-release fertilizer (Osmocote Plus; The Scotts Company LLC, Marysville, OH) that was mixed with the soil using a cement mixer with 88.7 g of fertilizer per pot before planting. Insect pests were controlled using standard recommended insecticides as necessary. Cyclanilide (0.011 mL a.i./L) + mepiquat chloride (0.045 mL a.i./L) (Stance; Bayer CropSciences, Research Triangle Park, NC, U.S.A) was applied with a backpack sprayer to manage cotton vegetative growth.

### 4.3. Experiment Layout

Each table in the greenhouse held one replication with one row of plants where the soil had no *V. dahliae* and one row of plants where the soil was infested with *V. dahliae* microsclerotia. The six genotypes were randomized down a row within inoculum levels (with or without *V. dahliae)*. The side of the table with or without *V. dahliae* was also randomized between replications. There were six replications. The test was repeated three times. The tests were conducted from 30 May to 28 October 2018, 25 April to 2 November 2019, and 27 June to 11 December 2019.

### 4.4. Disease Ratings

Plants were evaluated for Verticillium wilt symptoms starting 30–40 DAP, which was when the earliest symptoms developed on some plants. The day that first symptoms were seen on individual plants was recorded. Plants were rated for foliar disease severity based on the 0–5 scale as described by [36,37]. The 0–5 scale defines foliar symptoms as follows: 0 = 0%, no disease symptoms; 1 = 1–25%, minimum chlorosis at lower leaves; 2 = 26–50%, plant with chlorosis on lower and middle leaves; 3 = 51–75%, plant shows well developed symptoms of chlorotic, necrotic, and twisted terminal leaflets on one or more branches; 4 = >75%, more than three leaves show severe symptoms of chlorosis/necrosis; 5 = 100%, a complete plant death. At the end of each experiment, the stem was evaluated for discoloration. The rating for vascular cross-section discoloration (0–4 scale) was assigned according to [38] as follows: 0 = 0% discoloration or no disease symptoms; 1 = 25% of the cross-section shows discoloration; 2 = 26–50% of the cross-section has turned brown; 3 = 51–75% of the cross-section has turned brown; 4 = ≥76% of the cross-section has turned brown.

### 4.5. Physiological Measurements

Starting at 30 DAP, the rate of photosynthesis, stomatal conductance, intercellular carbon dioxide (Ci), and transpiration rate were measured. All physiological traits were repeatedly measured using the LI-6400 portable photosynthesis system (LI-6400XT; Li-Cor, Lincoln NE, USA) by choosing the youngest fully expanded mainstem leaves of an individual plant treated with and without *V. dahliae*. The light was provided by an integrated LED head (293 LED, LI-6400-02B). Within the cuvette, the air temperature was 28 °C, the flow rate was 400 µmoL s^−1^, and CO_2_ was maintained at 400 µmoL moL^−1^. The physiological trait measurements were collected at different growth stages of upland cotton including pre-flowering, peak flowering, boll setting, and boll cracking.

### 4.6. Yield and Fiber Quality

Bolls were hand-harvested at different positions and the seed cotton was ginned with a 10-saw laboratory-scale gin with no lint cleaner. Bolls harvested from different positions in the plant were used to calculate the seed cotton weight, seed weight, and lint production per plant. It should be noted that plants affected by Verticillium wilt produced bolls only at first and/or second position. For that reason, the fiber quality of bolls harvested from the first and second positions within the plant was tested with the high volume instrument (HVI) and advanced fiber information system (AFIS) at Texas Tech University, Fiber and Biopolymer Research Institute (FBRI), Lubbock, TX. For AFIS testing, three replicated blended samples with 3000 fibers were analyzed per sample. For sample preparation, a 0.50-g tuft of fibers was drawn into a 25-cm length sliver, and 9000 fibers were measured from each sample. FBRI performed testing of the cotton sample under constant climate-controlled conditions. The standard temperature for fiber property testing is 20 ± 2 °C at 65 ± 2% relative humidity. Before testing, samples were arranged in single layers and allowed to equilibrate for 48 h under standard atmospheric conditions. To minimize experimental error, the same technician ran all the samples in each year. The cotton samples were tested on the HVI, with four micronaire and ten length and strength measurements under standard laboratory conditions.

### 4.7. Statistical Analysis

Physiological traits such as photosynthesis rate, internal carbon concentration, and transpiration rate were measured repeatedly (12 times) throughout the cotton growth stages. Data for physiological traits were analyzed with SAS 9.4 (SAS Institute Inc) using the PROC MIXED (mixed model) procedure. Genotype, treatment (inoculated vs. healthy), and DAP were considered as independent variables. Plant_ID, which is a combination of replication, test, and treatment groups, was used as a random variable accounting for the repeated measurements of the same plant at different time points. The PROC MIXED procedure was also used to analyze disease severity ratings (DS). As the distribution of disease severity during the early stages of plant growth was very low, the last two severity ratings (during boll setting and harvesting period) were used for this data analysis. Genotype and growth stage were considered as independent variables, while Plant_ID was considered as a random variable. Shapiro–Wilk’s, Brown–Forsythe’s, and Levene’s tests were used for validating normality and homoscedasticity of all measured variables. When the data met the criteria for normality and homoscedasticity assumptions, data of all fiber quality traits, initial Verticillium wilt development (WILT_i_), vascular cross-section severity disease rate (VCDS), the total number of bolls per plant, and boll size were analyzed using general linear models (SAS PROC GLM). For non-normally distributed data, log transformation was applied. Tukey’s HSD test was used to determine differences among genotypes for different traits at the *p* = 0.05 level of significance. Mean comparison was performed using Genomics 6 (JMP, 2013).

## Figures and Tables

**Figure 1 plants-09-00857-f001:**
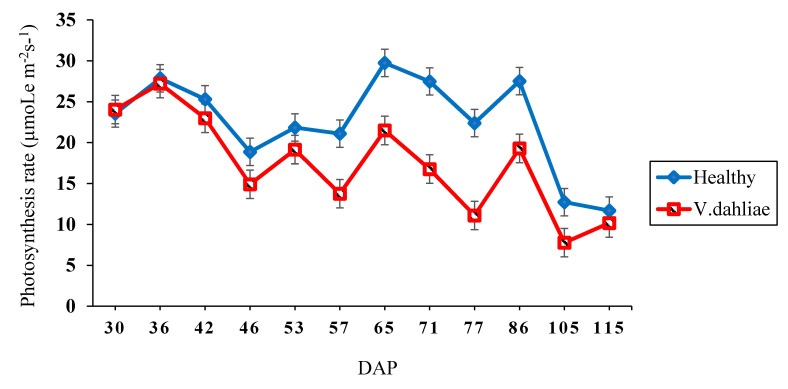
Photosynthesis rate over time of healthy and *Verticillium dahliae* inoculated greenhouse-grown upland cotton breeding lines (16-13-203V, 16-13-601V, 17-17-206V, and 17-17-606V) and cultivars (DP 1612 B2XF and FM 2484B2F), Each line graph represents the average photosynthesis rate for six genotypes, six replications, and three tests for healthy (blue) and *V. dahliae* treated (red) plants. The standard error bars shown for each means.

**Figure 2 plants-09-00857-f002:**
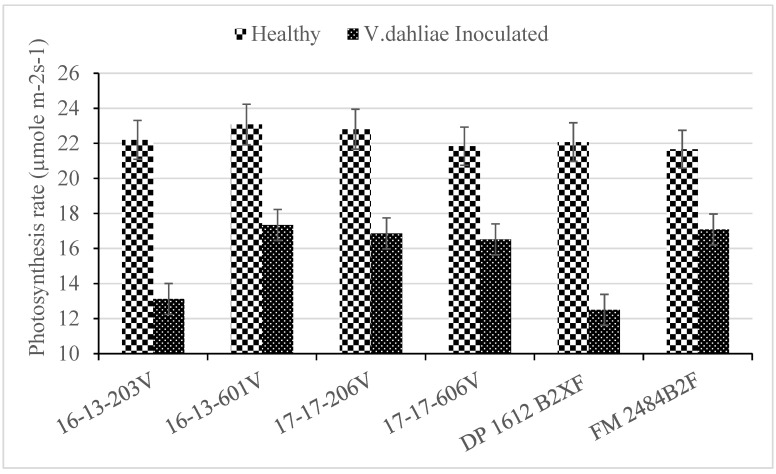
Impact of Verticillium wilt on photosynthesis rate of greenhouse-grown upland cotton breeding lines (16-13-203V, 16-13-601V, 17-17-206V, and 17-17-606V) and cultivars (DP 1612 B2XF and FM 2484B2F). The graph illustrates the average photosynthesis rate for 12 repeated measurements, 6 replications, and 3 tests for healthy and *V. dahliae* treated plants. Standard error bars are shown for each mean.

**Figure 3 plants-09-00857-f003:**
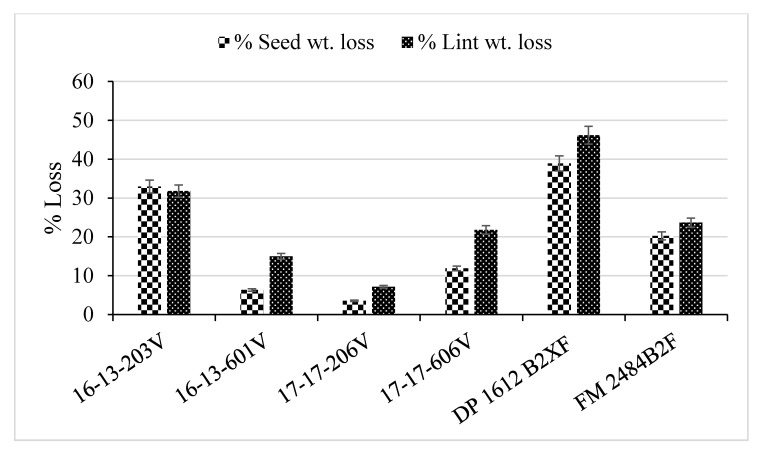
Percent (%) seed and lint yield loss of greenhouse-grown upland cotton breeding lines (16-13-203V, 16-13-601V, 17-17-206V, and 17-17-606V) and cultivars (DP 1612 B2XF and FM 2484B2F) due to artificial inoculation with *Verticillium dahliae*. The graph illustrates the average yield loss under Verticillium wilt pressure. Standard error bars are shown for each mean.

**Figure 4 plants-09-00857-f004:**
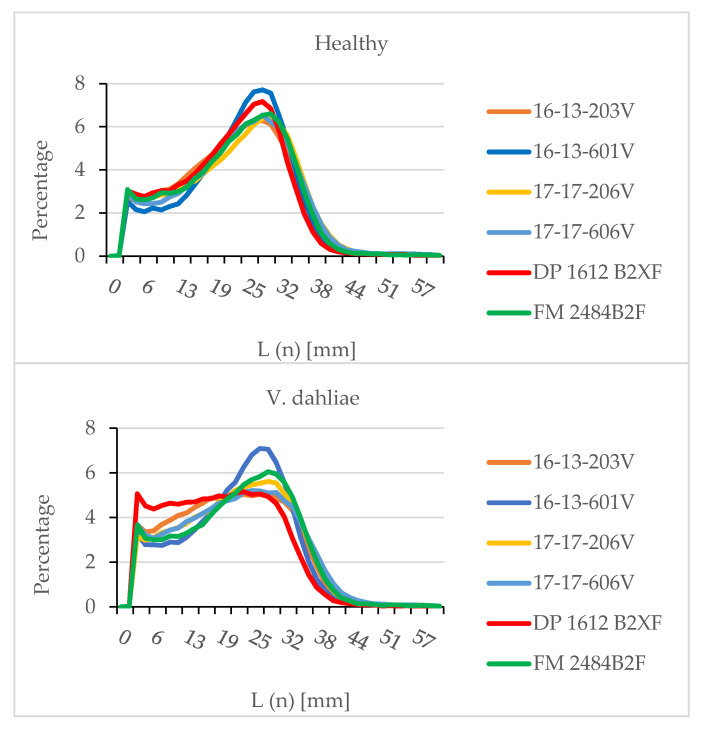
Fiber length distribution of greenhouse-grown upland cotton: healthy (**top**) and inoculated with *Verticillium dahliae* (**bottom**). L (n) [mm], length by number in millimeters.

**Figure 5 plants-09-00857-f005:**
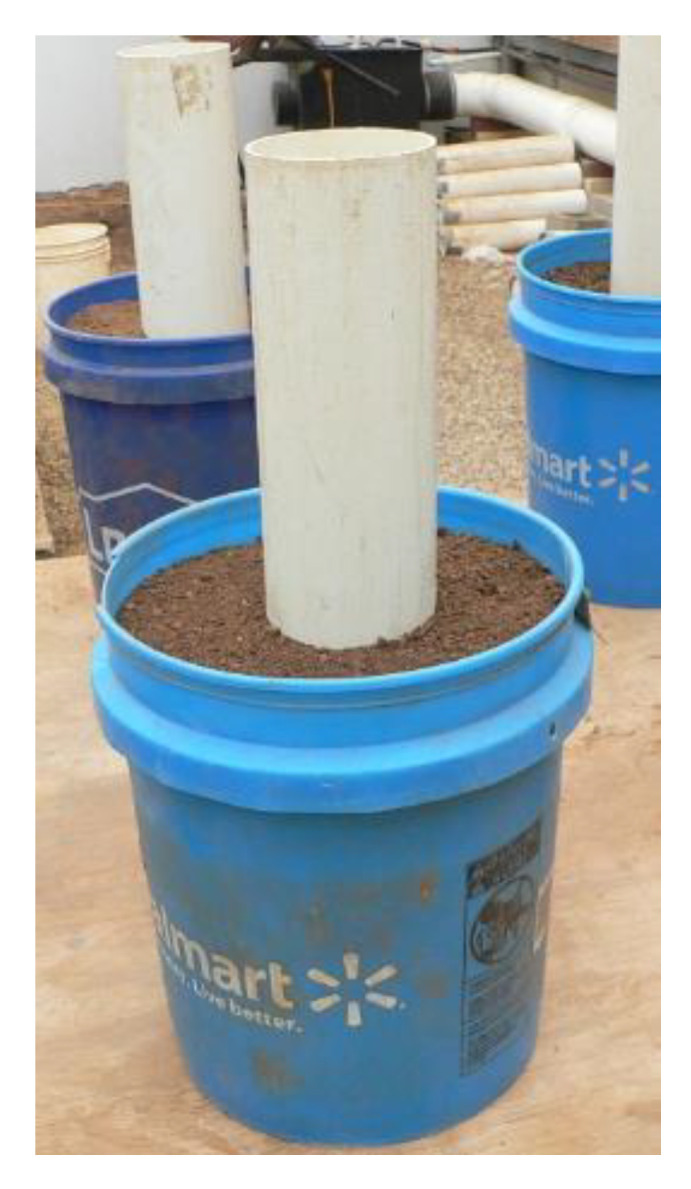
Setup for adding *Verticillium dahliae* inoculated soil to pots to measure impact on cotton photosynthesis rate, lint production, and fiber quality.

**Table 1 plants-09-00857-t001:** Least square means for initial Verticillium wilt development (WILT_i_), disease severity (DS), vascular cross-section severity disease rate (VCDS), photosynthesis rate, lint yield per plant (Lint) and seed yield per plant (Seed).

Genotype	WILT_i_ ^b^ (DAP)	DS ^c^	VCDS ^d^	Photosynthesis (µmoL m^−2^ s^−1^)	Lint (g)	Seed (g)
16-13-203V	44 b ^a^	3.7 a	2.2 ab	19.6 b	37 bc	53 bc
16-13-601V	67 a	2.0 b	1.0 c	22.0 a	42 ab	57 ab
17-17-206V	56 ab	2.6 b	1.4 c	22.2 a	45 a	62 a
17-17-606V	53 ab	2.0 b	1.5 bc	21.7 a	41 abc	54 bc
DP 1612 B2XF	55 ab	3.5 a	2.4 a	19.2 b	36 c	48 c
FM 2484B2F	47 b	2.3 b	1.0 c	21.8 a	45 a	60 ab

^a^ Means followed by the same letter within a column are not significantly different at the *p* < 0.05 probability level. ^b^ WILT_i_ is the time of initial Verticillium wilt development, given as days after planting (DAP). ^c^ Disease severity is on a 0 to 5 scale, where 0 = no disease and 5 = dead plant. ^d^ VCDS is on a 0 to 4 scale, where 0 = no vascular necrosis and 4 = 100% vascular necrosis. Tukey’s HSD test was used to determine differences among genotypes for different traits at the *p* < 0.05 level of significance.

**Table 2 plants-09-00857-t002:** Analysis of variance for photosynthesis rate, intercellular carbon dioxide concentration (Ci), and transpiration rate.

Source	Mean Squares			
	DF	Photosynthesis Rate (µmoL m^−2^ s^−1^)	Ci (µmoL CO_2_ moL^−1^)	Transpiration Rate (mmoL H_2_O m^−2^ s^−1^)
Inoculum ^b^ (+/−)	1	209.89 ***^a^	2.70 *	96.62 ***
Genotype	5	5.97 ***	2.80	2.61 **
Genotype × Inoculum	5	4.97 **	3.76	3.39 **
DAP	13	52.12 **	19.75 **	242.46 **
Inoculum × DAP	13	8.51 **	1.62	17.73 **
Genotype × DAP	65	0.98	1.88 **	1.17
Inoculum × Genotype × DAP	65	0.91	1.10	1.01
Error	854	41	3548	7.5

* Significant difference at *p* < 0.05, ** significant difference at *p* < 0.01, *** significant difference at *p* < 0.001. The inoculum was *Verticillium dahliae* microsclerotia mixed into the soil before planting (+) or absence of *V. dahliae* (−).

**Table 3 plants-09-00857-t003:** Least square means for fiber properties of micronaire, neps, short fiber content by number (SFC (n))), fine(mTex), immature fiber content (IFC), and maturity ratio (MR) with and without Verticillium wilt.

No Verticillium Wilt						
Genotypes	Mic	Neps	SFC (*n*)	Fine (mTex)	IFC	MR
16-13-203V	4.5 b ^a^	117 a	19.1 a	167 c	6.2 b	0.92 a
16-13-601V	4.1 c	156 a	15.7 c	152 d	7.0 a	0.92 a
17-17-206V	4.6 b	118 a	19.3 a	174 ab	6.0 b	0.91 ab
17-17-606V	3.9 c	166 a	18.6 a	153 d	7.0 a	0.90 ab
DP 1612 B2XF	5.0 a	145 a	16.1 bc	178 a	5.9 b	0.90 bc
FM 2484B2F	4.4 b	136 a	17.7 ab	168 bc	5.9 b	0.90 bc
With Verticillium Wilt						
16-13-203V	3.4 bc	331 ab	25.3 ab	146 bc	7.8 ab	0.85 bc
16-13-601V	3.9 ab	210 b	18.9 c	153 abc	7.3 abc	0.88 ab
17-17-206V	4.0 a	243 b	22.3 bc	164 a	6.9 bc	0.88 ab
17-17-606V	3.4 c	258 ab	23.1 bc	143 c	7.9 ab	0.85 bc
DP 1612 B2XF	3.6 bc	565 a	29.3 a	144 c	8.3 a	0.83 c
FM 2484B2F	4.1 a	191 b	20.4 bc	159 ab	6.5 c	0.90 a

^a^ Means followed by the same letter within a column are not significantly different at the *p* < 0.05 probability level. Tukey’s HSD test was used to determine differences among genotypes for different fiber quality traits at the *p* < 0.05 level of significance.

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
