# Peer review of "Impacts of Verticillium Wilt on Photosynthesis Rate, Lint Production, and Fiber Quality of Greenhouse-Grown Cotton (*Gossypium hirsutum*)"

_plants, 2020, doi:10.3390/plants9070857_

Round 1

Reviewer 1 Report

The authors investigate the impacts of Verticillium wilt on photosynthesis rate, lint production, and fiber quality of two commercial and four breeding greenhouse-grown cotton genotypes. The effect of Verticillium wilt on fiber quality was stronger in genotypes that are more susceptible. One breeding line fiber quality was almost unaffected by the pathogen infection.

The work is well designed and the results interesting. However, there are some minor points that I would appreciate to be considered:

  1. Results section, 3.1. Disease ratings.
    1. The genotype 17-17-606V is not commented in the text.
    2. Line 56 DAP please define in the text.
  2. First figure is No 2 and the last No 1. Please renumber the figures according to the order they appear.
  3. In the Discussion section, lines 161-162 the authors comment, “The magnitude of damage caused by Verticillium wilt on cotton lint and seed yield depends on the phenotypical stage of the plant when first foliar symptoms occur in the crop [28]. Our results did not necessarily agree with this observation”. In fact, there is a discrepancy. I would appreciate a comment about it. So, according to the authors. What could be the reason for that discrepancy?

Author Response

The genotype 17-17-606V is not commented in the text.

Response: We appreciate for pointing out that genotype 17-17-606V was not commented. We have included in the text.

  1. Line 56 DAP please define in the text

Response – We defined DAP as days after planting in the text

  1. First figure is No 2 and the last No 1. Please renumber the figures according to the order they appear.

Response: Thank you, we have corrected numbering of the figures.

  1. In the Discussion section, lines 161-162 the authors comment, “The magnitude of damage caused by Verticillium wilt on cotton lint and seed yield depends on the phenotypical stage of the plant when first foliar symptoms occur in the crop [28]. Our results did not necessarily agree with this observation”. In fact, there is a discrepancy. I would appreciate a comment about it. So, according to the authors. What could be the reason for that discrepancy?

Response: This may be because of the differences in the genotypes the current research and previous used. It seems some genotypes have different pathway or resistance mechanism to disease. For example, FM 2484B2F is our partial resistant check. It shows diseases symptom as early as the susceptible genotype 16-13-203V. However, FM 2484B2F tends to recover from the disease and maintain better yield under disease pressure. Therefore, although our resistant check show early first foliar symptoms, the lint yield per plant is still maintained.

Reviewer 2 Report

The manuscript reports interesting data about the behaviour of new cotton genotypes toward the soil borne pathogen Verticillium dahliae. The topic itself and the methodologies are not new and many other paper have been published on the same subject.  However, the manuscript could have prepared better for submission than it was. There are several points that need to improved before pubblication. Details are reported in the annotated manuscript and to facilitate the work of the Authors, all the comments have been summarized in a separate pdf file.

Among the different points annotated in the manuscript, the Authors must conisder carefully the aspects related the experimental design and the statistical analysis. Detail about the experiments are necessary and the Authors must clarify how many times they repetead the experiments (twice, thrice... it is not clear) as this point pose doubts abouth the soundness of the statistical analysis.

The "Discussion" section need to be edited, as many sentences refer to data that are not provided in the current manuscript. Many inferences refer to data available in literature, but they do not derive from the results described in this manuscript. The entire section need a tight editing work.

Many other point are reported in the annotated manuscript.

Author Response

Page 1

  1. The section needs to be improved, as it appears as a group of short and disconnected sentences, without a common thread.

Response: We have made some edits to improve the introduction flow. Thank you.

  1. Why do The Authors say "eventually"? Do they know alternate mechanisms of infections by V. dahliae in the soil?

Response: We appreciate your point. We removed the word ‘eventually’

  1. The citation of V. albo-atrum is not appropriate when talking about long living structures such as microsclerotia.

Response: We kept the citation as it was because the citation was referring to V. dahliae, but in the older literature, it was often called V. albo-atrum microsclerotia type.

Page 2

  1. The meaning of this acronym must be specified the first time used.

Response: The DAP Acronym is specified. Thank you.

  1. Why do the Authors use and compare the "least square means"? Is this due to the unequal number of observations for each combination of treatments? Anyway, for a better understanding the "actual values" of the different parameters must be reported.

Otherwise it is difficult to understand what are 44 "least square means" DAP....

Moreover, the type of test used to compare the means in each column, must be specified

Response: Least square means are means calculated after using a model or after ANOVA. As pointed out by the reviewer, we have a few missing data. Instead of arithmetic value, we must use least-square means that are adjusted for other terms in the model (e.g. covariates) and are less sensitive to missing data. For example, 16-13-203V genotype develops disease 44 days after planting for the first time. So, 44 is the standardized means over replication, treatments, and different tests for that genotype. The same thing applies to fiber quality analysis.

  1. The test used to compare the means must be specified.

Response: In the material and method section, we have mentioned that Tukey's HSD test was used to compare the means. For the convenience of the reader, we put the description under each table.

  1. The type of factorial experimental design must be specified.

Response: We have included more description in material and method section, under the experimental layout.

  1. The meaning of this interaction in terms of disease reduction per each genotype must be specified.

Response: We have included more explanation in the result section to make it clear. We have also explained this concept in the discussion section.

-When there is significant genotype x inoculum it means that some genotype(s) performed well in the absence of disease, while the same genotype (s) poorly performed under Verticillium wilt pressure.

Page 3

  1. The Authors must specify the experimental design they used, as it is not clear how they've got 854 DF. It the experiments were arranged according to a "factorial split plot" design, details must be added in the specific section "Statistical analysis". Otherwise, it is not possible to understand why and how the Authors pooled together all the data from all the different experiments

            Response: The degree of freedom of error is relatively high because the analysis of variance was performed by combining all three tests. Since physiological traits were measured repeatedly (12 times) as indicated in figure 1, the SAS PROC MIXED (mixed model) procedure was used where Plant ID that is the combination of replication, test, and treatment groups was used as a random factor. To analyze this data, we have also communicated with a well-experienced statistician.

  1. The meaning of the bars per each point must be specified. If the overall data from the different experiments have been pooled together in this graph, how has been calculated the standard error/deviation?

Response: This graph is the average of all replication, genotype, and inoculum levels. The standard error (SE) bar was created using Microsoft Excel. We have compared genotypes in the text using SE bars.

Page 4

  1. The x-axis must be visible

Response: We have corrected the axes, thank you.

  1. The meaning of bars must be specified. Are the differences among the different genotypes statistically significant? How are compared the data in the graph?

Response: We wrote the comparison of genotype in the text, thank you.

Are the differences among the different genotypes statistically significant?

Response: Yes, genotypes treated with the disease show significant differences, while there is no significant difference between healthy plants for photosynthesis rate.

How are compared the data in the graph?

Response: When there is no overlapping between standard error bars, genotypes are significantly different. The graph demonstrates, there no significant differences among healthy plants. While differences were observed among diseased plants.

The x-axis must be visible

Response: We have corrected the axes, thank you.

  1. What is the meaning of the bars on the top of the mean value? is it the standard error (n=?). Why the comparison by the Tukey's HSD test is not reported?

Response: In all bar graphs, we used Standard error (n=144 for photosynthesis) and 36 for yield loss. We prefer a standard error bar instead of Tukey’s HSD test. We used Tukey’s HSD test in the tables.

Page 5

  1. How do the parameters interact among each other? What is the meaning of this interaction? How the interaction affected the measured parameters?

Response: Genotype and Inoculum are independent variables in this study. The interaction effect between the genotype x inoculum tells us the behaviors of genotypes. For example, the lowest average short fiber content was recorded for DP 1612 B2XF in the absence of Verticillium wilt, while the highest average short fiber content was recorded for the same cultivar in the presence of Verticillium wilt.

  1. Why do the Authors use the "least square means" for the fiber properties instead of the specific value? Moreover, if the objective of the manuscript is to compare resistant and susceptible cotton's genotype why did the Authors compare the genotypes without considering the test used in the comparison must be specified.

Response: We have addressed this question on page 2 question number 2.

Page 6

  1. Is it expressed as percentage, pure number or what?

Response: Percentage and Frequency are interchangeably used in publications. So, we keep the frequency as it was. In this case, frequency is more appropriate talking about the statistic fiber length by number L(n). Percentage may be used for the statistic fiber length by weight L(w), since long fibers weigh more than short fibers. However, the statistic fiber length by number is more predictive of cotton yarn spinning performance and quality which is why we use it here.

  1. This seems a continuous line and the 0.0 point does not fit the with the beginning of the new graph. Please, split the two graphs.

Response: Graphs are separated and re-constructed. We have split the graph, instead of side by side, we arranged it top to bottom.

  1. As the lines in the graph seem "continuous lines" between the two theses, it would be better to separate the two graphs. The existence of differences among the different cotton genotypes should be reported.

     Response: Graphs are separated, thank you.

  1. The "Discussion" section needs to be edited, as many sentences refer to data that are not provided in this manuscript. Many inferences refer to data available in literature, but they do not derive from the results described in this manuscript. The entire section needs a tight editing work.

Response: We have made modifications to the discussion section, thank you.

  1. A huge number of papers report information about the mechanisms of resistance of cotton towards V. dahliae, and the Authors must detail the most relevant published recently.

Response: To utilize all the cited articles, we have made some additions in the text and added 2017 reference [28] to update relevant research.

Page 7

  1. A table would facilitate better reading of the different genotypes used in the experiments. Here, it would be better to clarify shortly how the breeding lines have been obtained, specifying the criteria used for the breeding or citing the references in which these data are available.

Response: These breeding lines were developed in our breeding program. So, we have included more descriptions into the text, under the material and method in the germplasm section.

  1. How was it ascertained: defoliating type isolate?

Response: Five isolates of V. dahliae that were taken from five regionally separate cotton fields exhibiting typical defoliation symptoms were sent to Dr. S. Grung and characterized as defoliating types as part of this publication (Hu, X-P, Gurung, S., Short, D. P. G., Sandoya, G. V., Shang, W-J, Hayes, R. J., Davis, R. M., and Subbarao, K. V. 2015. Nondefoliating and defoliating strains from cotton correlate with races 1 and 2 of Verticillium dahliae. Plant Disease 99:1713-1720. http://dx.doi.org/10.1094/PDIS-03-15-0261-RE). It is assumed that Verticillium dahliae isolated from cotton exhibiting substantial defoliation in this region, is the defoliating type.

  1. What was the sieve's mesh?

Response: The soil was sieved with a 0.635 cm mesh sieve (added).

  1. How it was ascertained? The Authors must specify the methods or the reference to their previous work, if any.

Response: They were run by A&L Plains Agricultural Laboratory, Lubbock, TX

  1. The acronym is not immediate for non-expert reader; please specify "organic matter"

Response: correction made as suggested, thank you.

  1. Was this value different from that reported below (808 microsclerotia/ g soil)?

Response: the desired density was 808 microsclerotia/ g soil

Page 8

  1. Twice? It seems thrice... What is wrong here?

Response: Corrected to three times, thank you.

  1. Do the Authors mean six plants per each genotype?

Response: Yes, genotypes were replicated 6 times in each inoculum.

  1. The meaning of this acronym must be specified the first time is used.

Response: The acronym is specified.

  1. Considering the diameter of the stem, the Authors must add some details about the procedures they used to rate the vascular cross-section discoloration according to citation 16. Do the Authors used a stereo-microscope?

Response: We have not used any microscope to rate vascular cross-section discoloration. The vascular discoloration caused by Verticillium wilt is clear and visible. So, we use visual rating following the standard scale.

Page 9

  1. What was the EXPERIMENTAL DESIGN? Details must be added.

Response: The Experimental design was detailed under the experimental layout section in the material and method. It was a factorial arrangement of 6 genotypes with replicated 6, completely randomized in each inoculum level. The test was conducted three times.

  1. That is.... how many times?

Response: We measure photosynthesis rate 12 times.

  1. Do this sentence imply that the data from the three different experiments have been elaborated together?

Response: Yes, all three tests were combined and analyzed.